# Genetic Analysis of the Full-Length *gag* Gene from the Earliest Korean Subclade B of HIV-1: An Outbreak among Korean Hemophiliacs

**DOI:** 10.3390/v11060545

**Published:** 2019-06-11

**Authors:** Young-Keol Cho, Jung-Eun Kim, Brian T. Foley

**Affiliations:** 1Department of Microbiology, University of Ulsan College of Medicine, Seoul 05505, Korea; kimje2000@nate.com; 2HIV Databases, Theoretical Biology and Biophysics Group, Los Alamos National Laboratory, Los Alamos, NM 87544, USA; btf@lanl.gov

**Keywords:** HIV-1 *gag* gene, Korean subclade of subtype B, phylogenetic analysis, signature pattern, hemophiliacs

## Abstract

We determined the earliest full-length HIV-1 *gag* gene sequences in 110 patients with HIV-1, including 20 hemophiliacs (HPs) and 90 local controls (LCs). The *gag* gene from stored sera was amplified using RT-PCR, and was subjected to direct sequencing. Phylogenetic analysis indicated that 94 and 16 sequences belonged to the Korean subclade of HIV-1 subtype B (KSB) and subtype B, respectively. A total of 12 signature pattern amino acids were found within the KSB, distinct from the worldwide consensus of subtype B. Within the KSB, the *gag* gene sequences from donors O and P and those from the 20 HPs comprised two subclusters. In particular, sequences from donor O strongly clustered with those of eight HPs. Moreover, signature pattern analysis indicated that 14 signature nucleotides were shared between the HPs and LCs within KSB (*p* < 0.01). Among the 14 nucleotides, positions 9 and 5 belonged to clusters O and P, respectively. In conclusion, signature pattern analysis for the *gag* gene revealed 12 signature pattern residues within the KSB and also confirmed the previous conclusion that the 20 HPs were infected with viruses due to incompletely inactivated clotting factor IX. This study is the first genetic analysis of the HIV-1 *gag* gene in Korea.

## 1. Introduction

We have previously conducted a nationwide genetic analysis of HIV-1 with sera from individuals in the early stages of infection with HIV-1 (before 1994) with the aim of identifying the cause of an HIV-1 outbreak among hemophiliacs in Korea in 1990–1994. As a result of these molecular epidemiological studies, viruses from two plasma donors, who were paid to donate, were identified as the agents of infection of 20 HIV-1-infected hemophiliacs [1,2,3,4,5]. In these studies [1,2,3,4,5], we conducted in-depth analysis of the *pol, vif, env*, and *nef* genes in the Korean subclade of subtype B (KSB) [6,7], but the *gag* gene is yet to be studied in Korea.

In the study reported here, we tried to obtain the earliest available full-length *gag* sequences and identify signature pattern residues in the *gag* gene of KSB, which are distinct from the worldwide consensus subtype B.

KSB is presumed to have originated from strains in the USA through the founder effect [6,7], and the most recent common ancestor is estimated to have been active around 1984 [8]. The earliest case KSB HIV-1, was diagnosed in 1988 [2,3,4,5].

Signature pattern analysis is a method for assessing viral sequence relatedness in viral epidemiology. This method is based on the identification of particular sites in amino acid or nucleic acid alignments of variable sequences that are distinct to a query set or are representative of a query set of sequences which are rare in a background set in their respective positions in an alignment [9].

In addition to the signature pattern analysis [9,10] of KSB versus subtype B, we analyzed the cause of an HIV-1 outbreak among 20 hemophilia B patients (HPs B) [1,2,3,4]. There were outbreaks of hepatitis A virus infection in 1998–1999 [11], and hepatitis C virus (HCV) infection in individuals who used the same products [1]. The Supreme Court ruled that the manufacturer was responsible for the transmission of HIV-1 and HCV in a single cohort (Case No. 2008 Da16776 and 2013 Da26708).

A global HIV-1 vaccine will need to elicit immune responses capable of providing protection against a wide range of HIV-1 variants. In particular, effector Gag-specific T lymphocytes are very important for virologic control. One approach is to design vaccines specific to each geographic region, involving antigens which are tailor-made to combat local circulating HIV-1 strains [12]. Using this approach, in order to develop a vaccine to be used in Korea in the future, it is essential to identify the nucleotide sequence of the *gag* gene that was first introduced into Korea. However, no studies have been conducted into the KSB *gag* gene, although the *pol* and *env* structural genes of HIV-1 have been analyzed.

This study is the final in a series of genetic analyses of KSB, and provides a comprehensive nationwide analysis of the earliest-known *gag* sequences in Korea [2,3,4,5]. This is the first study to compare several genes in an HIV-1 outbreak with transmission. We identified 12 signature amino acid residue patterns in the KSB Gag protein that are distinct from worldwide consensus B, and confirmed a previously postulated epidemiological link between the viruses that infected 20 HPs and two plasma donors, in contrast with those that infected local controls (LCs), as shown in previous studies [2,3,4,5].

## 2. Materials and Methods

### 2.1. Study Patients and Samples

Three HIV-1-infected plasma donors were diagnosed during the primary infection in 1990–1992. Their plasma was used to manufacture clotting factor 9. The details have been described previously [2,3,4,5]. In total, 251 HIV-1-infected patients were diagnosed in Korea before 1992. Based on epidemiological data, it appears that 105 patients were domestically infected with KSB, the transmission of which began in 1988 [1,2,13,14], and 40 patients were infected with Western subtype B. There were no samples from about 20 people, and researchers could therefore not determine their HIV-1 subtype. In the present study, *gag* genes were sequenced from 94 KSB-infected patients, including three plasma donors and 20 hemophiliacs (designated HPs 1–20). A total of 83 patients were diagnosed between 1988 and 1993, and 11 patients were diagnosed after 1993 [2,3,4]. The dataset included 16 subtype B-infected patients, including HPs 21–23 and 13 patients (*n* = 110). The subtype B virus was derived mainly from the United States of America, with one case originating in Iran. The carriers appear to have been five prostitutes from the United States Forces Korea, three hemophiliacs infected in the USA and Iran, an overseas sailor and the wife of an overseas sailor, four homosexual men, and a heterosexual man who visited the USA.

The earliest sera used in this study were collected between April 1990 and April 1993 from nearly all of the patients newly diagnosed with HIV-1 infection at the division of AIDS of the Korean NIH [2]. Samples from four HPs (9, 15, 19, and 20) were obtained in 2002.

Informed written consent was obtained from all living study patients. This study was approved by the institutional review board of the Asan Medical Center (Code 2012-0390, 4 June 2012).

### 2.2. RNA Preparation and *gag* Gene Amplification

Total RNA was extracted from 300 μL serum samples using a QIAamp UltraSens Viral RNA kit (Qiagen, Hilden, Germany), as described previously [2,3,4,5]. RNA was reverse transcribed using Superscript III reverse transcriptase (Invitrogen, Waltham, MA, USA) and the gene-specific primer 524 for the terminal 5ʹ LTR and the full *gag* gene (5′-CATTGTTTAACTTTTGGGCCATCC-3′). The *gag* gene of KSB was amplified using nested PCR with TaKaRa r-Taq (Takara Bio Inc., Shiga, Japan). The first and second PCR reactions were performed in 20 and 50 μL reaction mixtures, respectively. The outer primer pairs were 503k (5′-CCKTCTgTTGTGTGACTCTGGTAA-3′) and 524 (5′-CATTGTTTAACTTTTGGGCCATCC-3′), and the inner primer pairs were 504F (5′-TCTCTAGCAGTGGCGCCCGAAC-3′) or 505 (5′-GAGACATGGGTGCGAGAGCGT-3′) and 522 (5′-ACTGTCCTACTTTGATAAAACCTC-3′), respectively. After initial denaturation at 95 °C for two minutes, 35 cycles were run under the following conditions: 95 °C for 30 s, 52 °C for 30 s, and 72 °C for 2 min 30 s, followed by a final extension step at 72 °C for 10 min. The second PCR was performed with 1 μL of the first PCR product, and the cycling conditions were as follows: 95 °C for 30 s, 55 °C for 30 s, and 72 °C for 2 min, with a final extension step at 72 °C for 10 min. For the amplification of subtype B the *gag* gene was amplified using nested PCR with TaKaRa Ex Taq (Takara Bio Inc., Shiga, Japan). The outer primer pairs were CE1 (5’-CGAGAGCTGCATCCGGAGTACTA-3’) and 524 (5′-CATTGTTTAACTTTTGGGCCATCC-3′), while the inner primer pairs were 504F or 501 (5’-GTGTGGCCTGGGCGGGACTG-3’) and 522 (5′-ACTGTCCTACTTTGATAAAACCTC-3′), respectively. The amplicons were directly sequenced using Applied Biosystems 3730XL.

### 2.3. Phylogenetic Tree Analysis

A total of 187 sequences were obtained from 20 HPs. Sequences from 74 LCs, and 16 subtype B-infected patients were aligned against the HIV-1 subtype reference set from the HIV Sequence Database (http://hiv-web.lanl.gov/content/hiv-db/Subtype_REF/align.html), and phylogenetic trees were then built using the DNAML Maximum Likelihood method, PAUP maximum parsimony, and PhyML programs [15]. The trees generated by each method, as well as by the general time-reversible model of evolution plus site-specific rate heterogeneity, produced the same donor O and donor P subclades, within KSB of the HIV-1 subtype B [5].

### 2.4. Viral Signature Pattern Analysis (VESPA)

The Viral Signature Pattern Analysis (VESPA) program (http://www.hiv.lanl.gov/content/sequence/VESPA/vespa.html) was used to identify sites within each sequence group that were distinct from the other groups [16].

### 2.5. Statistical Analysis

Data are presented as means ± standard deviation. Statistical significance was determined using Student’s two-tailed *t*-tests, Chi-square tests, and Fisher’s exact test using SPSS version 12.0. Results were deemed to be statistically significant when the *p* value was less than 0.05.

### 2.6. Nucleotide Sequence Data

The GenBank accession numbers for the sequences in this study were MK548700-MK548886, EF370220, EF370383, EU047616, EU047617, EU047637, EU047656, EU047669, JN636165, JN636182, and JN636293.

## 3. Results

### 3.1. Origin of the KSB of Subtype B

A major contribution of the current study is the inclusion of the sequences of the earliest KSB-infected patients (Appendix A). We failed to amplify the *gag* gene in only two (BGO diagnosed in 1988, and YJS in 1991) of all the KSB-infected patients diagnosed before 1993, though their *pol* and *vif* sequences have previously been obtained and analyzed [2,3,4].

### 3.2. Molecular Epidemiologic Data on the gag Gene

We analyzed 187 *gag* gene sequences from 110 patients (Appendix A). Phylogenetic analysis revealed that the earliest 96 sequences from 94 patients (three plasma donors, 20 HPs, and 71 LCs) belonged to the KSB, whereas 16 of the earliest sequences from 16 patients belonged to subtype B (Figure 1). The 96 KSB sequences were subdivided into several clusters, including two large clusters (‘O’, which comprised 10 sequences, and ‘P’, which comprised 11 sequences; Figure 1) that included 18 HPs and plasma donors O and P. In particular, the two sequences from donor O clustered closely with eight HPs (1–4, 6, 8, 10, and 18; cluster O). In contrast, the two sequences from donor P clustered with 10 HPs with an intervening sequence, leaving the sequences of HP 5 and HP 9 outside cluster P (Figure 1). Using different phylogenetic methods, including neighbor-joining, maximum likelihood and maximum parsimony, and modified data sets such as including other HIV-1 M group subtypes for outgroup, resulted in nearly identical tree topologies, especially in regard to the KSB subclade of subtype B and the O and P clusters within this KSB subclade.

### 3.3. Nucleotide/Amino Acid Signature Patterns

We analyzed the signature patterns of the *gag* gene sequences from 94 KSB-infected patients and compared them with those from 16 subtype B-infected patients. The assessment of signature pattern nucleotides in clusters O and P indicated that nine and five positions showed statistically significant difference in frequency compared with nucleotide sequences in 85 and 81 other KSB, respectively, including HPs (*p* < 0.01), along with a significant difference between the two clusters (*p* < 0.01) (Table 1). Among the 14 nucleotide positions, nucleotides at positions eight and two in clusters O and P, respectively, were perfectly conserved in all sequences of all of the hemophiliac patients of each cluster. Among these nine positions in cluster O, the “A” at nucleotide position 1897 was not detected in HP-6 (MK548714-715) and was detected in only one of five amplicons from donor O (KF561442) (Table 1; Appendix A). In cluster P, only five nucleotide positions were identified as being highly significant when compared with other KSB because of the long time lag between primary infection and sampling time (Table 1; Appendix A).

### 3.4. Korean Signature Pattern Amino Acid Residues and Duplication of the PTAP Motif in the p6 Gag Protein

In this study, we used the earliest samples from 94 KSB-infected patients. We found that the signature pattern amino acids at 12 residues were distinct from the worldwide consensus B (Table 2). Six (HJHi and HP-23 LJH in subtype B; KJS, KMH, HP-19, and HP-20 in KSB) of the 110 patients possessed a PT/SAP motif duplication in the p6 Gag protein. In patient KJS, the duplication was detected in June 1999, January 2002, and August 2004, although the protein was originally wild type in 1991. Interestingly, his wife’s sequences (KMH) were also wild type until April 2007, and then changed to a duplication of “PAP” in January 2008.

In HPs 19 and 20 (MK548736), the duplication was first detected in samples obtained nine and seven years after diagnosis, respectively, although the plasma donor’s sequences were wild type. Patient HJHi, diagnosed in 1987, was infected with subtype B. She was a long-term slow progressor (LTSP). She had a duplication of “PTAPEE” in both sequences obtained in 1991 (MK548860-61). Aside from patients HJHi and HP-23 (EF370375), the clinical courses of these patients are depicted in [17,18]. Briefly, patients KJS and KMH were also LTSPs. Four HPs (17, 19, 20, and 23) were typical progressors. HP-17 was initially wild type in 1993 (MK548732-33), but after 10 years, all sequences were identified as “PTAPPAESFR PTAPPAESFR”(KJ140262).

### 3.5. Comparison of the Frequency of 100%-Specific Nucleotides among the Nine Genes

The *vif, vpr, pol,* and *env* genes contained 100% specific nucleotides at 2, 1, 3, and 4 positions in clusters O and P [2,3,5], respectively, in contrast to 0% in LCs, whereas the *nef* gene [4] did not exhibit specific nucleotides at any position, even within cluster O, as a result of the large number of patients assessed. In the present study, there was only one specific nucleotide, at position 1899 (Table 3). In total, there were only 10 specific nucleotide differences between clusters O or P and LCs (Table 3) among the nine genes (about 8600 bp).

### 3.6. Sequence Identities of Donors O and P

The sequence similarity in both clusters increased in the following order: *pol*, *vif*, *nef,* and *gag*. In donor O, the intrapersonal sequence similarities between the two *gag* amplicons obtained in October 1991 and February 2001 were 99.97% and 98.54%. The sequence similarity in cluster O between donor O’s *gag* sequence in 1991 and each HP’s sequence averaged 98.94% ± 0.2%, which was the second highest among the five genes. In donor P, the intrapersonal sequence similarity between the two *gag* amplicons obtained in October 1993 was 99.87%. In cluster P, the sequence similarity between donor P’s *gag* sequence and each HP’s was a little lower (98.88% ± 0.5%) than in cluster O, primarily because the four samples from the HPs (nos. 9, 15, 19, and 20) obtained after 2002 exhibited the lowest sequence similarity (nucleotide identity 96.6–98.1%) (Table 4). When the five genes were compared, the sequence identity of the *gag* gene was closest to the *pol* gene (Table 4).

In particular, HPs 19 and 20 had 18-bp and 30-bp insertions, respectively, containing “PTAP” motif duplications compared with consensus sequences.

### 3.7. Correlation between Sampling Intervals and Number of Nucleotide Differences

We determined the correlation between the sampling intervals after the outbreak (around January 1990) and the number of nucleotide differences observed, relative to those of the corresponding plasma donor. The full-length *gag* gene sequences from donors O and P were 1503 bp. Four patients who were first sampled in 2002 displayed the highest difference (43.5 ± 16.8 nucleotides over 146 ± 16 months), with a nucleotide variation of 19 ± 14, or approximately 1.26% over 55 ± 48 months. In cluster O, nucleotide variation was 14 ± 3, or approximately 0.93% over 31 ± 8 months, whereas in cluster P, nucleotide variation was 23 ± 18, or approximately 1.53% over 71 ± 56 months. The overall correlation coefficient, γ, was estimated to be 0.89 (*p* < 0.0001) (Figure 2) and was higher than the correlation coefficients (0.74–0.84) for the *nef*, *env*, and *vif* genes [3,4,5]. When the PTAP motif duplications were excluded from HP-19 and HP-20, the correlation coefficient was even higher (0.93, *p* < 0.001).

## 4. Discussion

In this study, we determined the full-length *gag* sequences of the 110 earliest-known patients infected with HIV-1 subtype B and found 12 signature pattern amino acids for KSB compared to subtype B. We confirmed an epidemiological link between the viruses that infected 20 HPs and two plasma donors, in contrast to those that infected 85 LCs in the same way, as seen in previous studies [1,2,3,4,5]. Signature pattern analysis identified 14 signature nucleotides shared between HPs and LCs within KSB (*p* < 0.01). The signature pattern nucleotide, which was perfectly conserved in all sequences in clusters O and P, in contrast to none in LCs as seen in the *pol*, *vif,* and *env* genes, was observed at only one position, 1899 (Table 3) [5].

KSB is a distinct monophyletic clade within HIV-1 subtype B, and is unrelated to any of the international sequences recorded in the Los Alamos Database as of December 2010. KSB is presumed to have originated from strains in the USA through a founder effect [6,7]. In population genetics, the founder effect is the loss of genetic variation that occurs when a new population is established by a very small number of individuals from a larger population. As a result of the loss of genetic variation, the new population may be distinctively different from the parent population from which it is derived. In the current study, 12 signature amino acids of KSB were derived from the various amino acids from Western subtype B; in particular, seven of the 12 positions derived from minor amino acids, from non-KSB subtype B (Table 2). In previous studies, we identified the *pol* (*n* = 97), *vif* (*n* = 127), *env* (*n* = 77), and *nef* (*n* = 143) gene sequences within KSB [1,2,3,4,5] and we included the same patients in the current study.

In the phylogenetic tree analysis of the *gag* gene, as with the *nef* and *env* genes, cluster P could not be supported by bootstrap values greater than 75%. Only a few sites were found to link donors O and P to the HPs, and bootstrap re-sampling would erase this signal [2,3,4,5]. Hence, signature pattern analysis is more suitable than bootstrap value in phylogenetic tree analysis to obtain evidence of an epidemiological link. In cluster P, in particular, there were four samples with more than 10 years between primary infection and sampling time, and only 10 sequences out of 12 HPs were included. No discordant subtypes were observed, although there was a recombinant strain between KSB and subtype D in Korea [19].

It is known that the PTAP motif is necessary for viral packaging. The duplication of the PTAP motif in the p6 Gag protein enhances the replication fitness of HIV-1 C by engaging the Tsg101 host protein with a high affinity [20]. The frequency of PTAP duplication in this study was significantly lower (5.5%) than in subtype C (16% or 8/50, *p* < 0.05). However, in this study, no significant differences were observed in the frequency of PT/SAP duplication between KSB (4.3% or 4/94) and subtype B-infected patients (12.5% or 2/16). Recently, the importance of p6 Gag protein as a substrate of the insulin-degrading enzyme has been investigated [21].

The present study had some limitations. Firstly, sera sampled from different individuals at various time points were used, and the time lag between primary infection and sampling was particularly long in cluster P (Appendix A). Secondly, different numbers of samples were compared among the five genes. Thirdly, two patients out of all the KSB-infected patients diagnosed before 1993 in Korea were not included because of repeated failures of PCR amplification. However, despite the variable sampling time points and the possibility of super-infection by both donors, signature pattern nucleotides were fully conserved at 10 positions, with a rare conservation in LCs (Table 1).

Our results show identification of 12 signature amino acid residue patterns in the KSB Gag protein that are distinct from worldwide consensus subtype B and confirm a clear link between the domestic clotting factor and the 20 KSB-infected HPs.

## Figures and Tables

**Figure 1 viruses-11-00545-f001:**
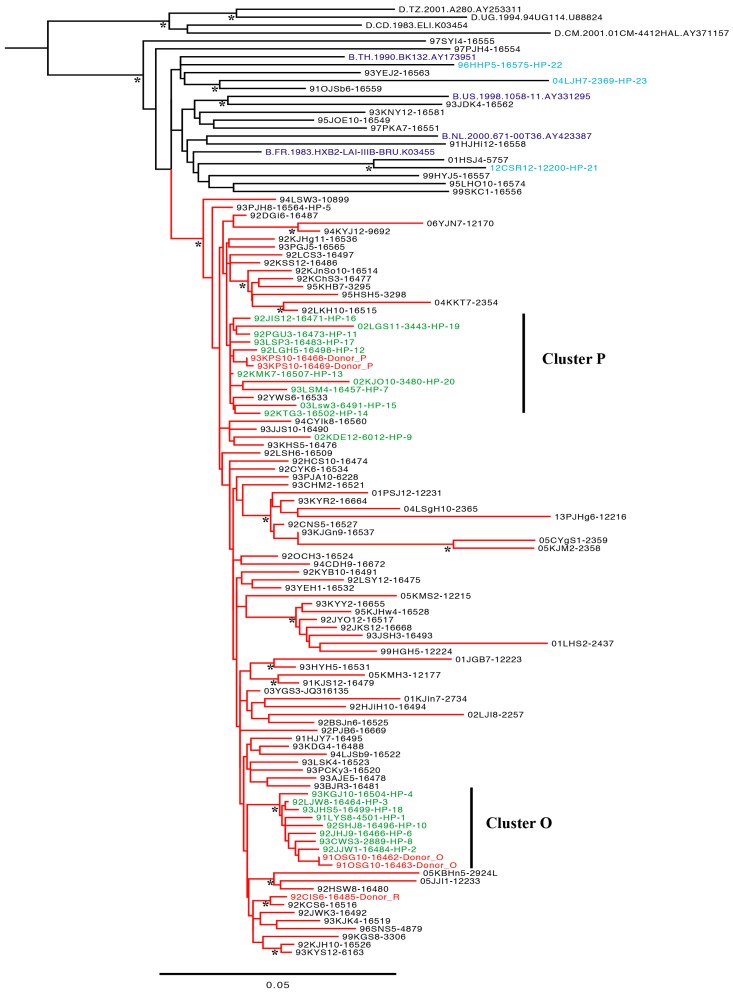
Phylogenetic tree of the earliest full-length *gag* gene sequences (1503 bp) of 110 Korean HIV-1 patients: 20 hemophiliacs (HPs); plasma donors O, P, and R; 71 local control patients infected with the Korean subclade of HIV-1 subtype B (KSB); and 16 non-KSB-infected patients (including three HPs, 21–23) infected with subtype B. Ninety-six sequences belonged to KSB (shown as a red horizontal line), and 16 subtype B Korean sequences and reference sequences are shown as a dark horizontal line. Nine patients, including donor O (Cluster O: 1–4, 6, 8, 10, and 18), strongly clustered within the KSB-infected local control patients. Sequences from donor P clustered with 10 HPs, leaving only two sequences—HPs 5 and 9—outside cluster P, although in more conserved full-length *pol* gene analysis, all 12 HPs clustered with donor P [2]. Three HPs (21–23) were infected by clotting factor purchased outside Korea (blue color). The two digits prior to patient IDs and the one or two digits after patient IDs denote the year and month of sampling, respectively. The reliability of the tree was evaluated by 1000 bootstrap replicates. The asterisks in the node show subclasses with more than 75% bootstrap support.

**Figure 2 viruses-11-00545-f002:**
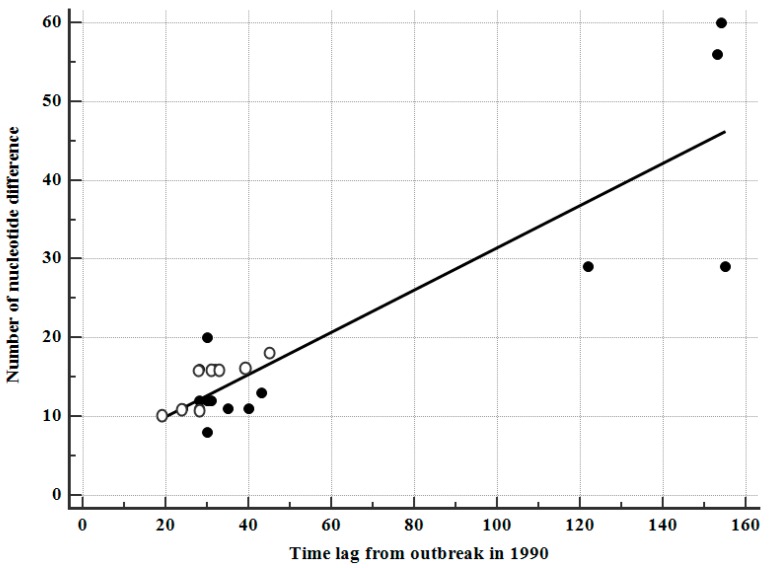
Correlation of the time lag (months) between the outbreak (around January 1990) and sampling and the number of nucleotide differences in the 20 HPs, compared to corresponding plasma donors. Four patients (HPs 9, 15, 19, and 20) who provided serum samples after 2002 exhibited the most differences (≥29). The correlation coefficient, γ, was 0.89 (*p* < 0.0001). Eight hemophiliacs infected with HIV-1 originated from plasma donor O and 12 hemophiliacs infected from plasma donor P are indicated with white and black circles, respectively.

**Table 1 viruses-11-00545-t001:** Frequency of the signature pattern nucleotides in the *gag* gene in 94 KSB-infected patients.

14 Signature Nucleotide Positions														
Nucleotide positions in HXB2	807	823	940	1012	1151	1264	1281	1674	1707	1722	1897	1899	2155	2190
Nucleotide in HXB2	A	A	T	C	A	A	C	C	C	G	G	A	A	A
Signature nucleotide	G						T	T	T	A	A	G	T	G
Cluster-O (*n* = 9)	1.0^a^						1.0	1.0	1.0	1.0	0.89	1.0	1.0	1.0
Other KSB (*n* = 85)	0.04^b^						0.13	0.05	0.11	0.05	0.02	0.00	0.13	0.13
Signature nucleotide		G	C	A	G	G								
Cluster-P (*n* = 13)		0.85	1.0	0.92	1.0	0.92								
Other KSB (*n* = 81)		0.25	0.04	0.12	0.30	0.40								

Cluster O contained “PSAP” in the “PTAP motif,” as it had a T instead of A at position 2155. Nucleotide positions in HXB2 indicate the position of the nucleotide in the second row. The fourth and seventh lines show the frequencies of these nine and five signature nucleotides among the hemophiliacs’ viral sequences and the fifth and eighth lines show their frequencies among the 85 and 81 background local control sequences (other KSB). 1^a^ and 0.04^b^ denote 100% and 4% of all sequences within cluster O and other KSB, respectively. We did not annotate values because there was no statistically significant difference between clusters O and P, and between each cluster and other KSBs.

**Table 2 viruses-11-00545-t002:** Signature pattern residues in the Gag proteins of 110 infected patients.

Signature Pattern Amino Acids at 12 Residues												
Position in HXB2	30	34	62	94	102	114	119	138	389	403	483	490
KSB (*n* = 94)	R_83_	L_78_	G_86_	I_89_	E_77_	K_93_	T_89_	L_93_	T_74_	K_89_	L_88_	R_84_
	k_11_	i_14_	e_5_	v_5_	d_14_	n_1_	a_5_	m_1_	i_10_	r_5_	p_3_	k_20_
		v_2_	a_2_, r_1_		k_1_, h_1_				p_10_		q_2_, k_2_, h_1_	
Non-KSB (*n* = 25)	k_13_	I_20_	g_15_	V_19_	e_15_	k_13_	A_24_	l_15_	i_9_	r_14_	l_13_	k_15_
	r_9_	l_4_	e_6_	i_5_	d_10_	r_7_	t_1_	m_5_	t_6_	k_11_	m_9_	r_10_
	q_3_		i_2_			t_4_		i_3_, v_2_	n_3_, s_3_		r_2_	
			v1			n_1_			m_3_, g_1_		q_1_	

Twelve residues were found to be signature pattern residues based on the criteria (>75%) applied in a previous study [2]. Signature amino acids (capital letters) in all 12 sites of the KSB were derived from the various amino acids of the non-KSB subtype B. Five positions (62, 102, 114, 138, and 483) were derived from major amino acids, and the remaining seven positions were derived from minor amino acids from non-KSB subtype B. This finding provides evidence for a founder effect, because KSB was derived from its parent, subtype B [1,2].

**Table 3 viruses-11-00545-t003:** Comparison of the frequency of the signature pattern nucleotides among six genes.

10 Specific Nucleotides												
Gene	*gag*	*pol*			*vif*	*vpr*	*tat, rev, vpu*	*env*				*nef*
No. of KSB (*n*)	94	91 [2]			127 [3]	52*	52*	77 [5]				143 [4]
Nucleotide position	1899	2321	4235	5070	5253	5621	None	6473	8252	8670	8774	None
Signature	G	G		A	A				T	C	T	
Cluster-O (*n* = 9)	1.0	1.0		1.0	1.0				1.0	1.0	1.0	
Other KSB (*n* = 85)	0.0	0.0		0.0	0.0				0.0	0.0	0.0	
Signature			G			A		G				
Cluster-P (*n* = 13)			1.0			1.0		1.0				
Other KSB (*n* = 81)			0.0			0.0		0.0				

*n* = the number of patients infected with KSB, including 20 hemophiliacs and local controls. Position 5070 also belongs to the *vif* gene. Nucleotide positions in the third row indicate the position of the nucleotides in the fourth and seventh rows. The fifth and eighth lines show the frequencies of these seven and three signature nucleotides among the hemophiliacs’ viral sequences and the sixth and ninth lines show their frequencies among the 85 and 81 background local control sequences (other KSB). 1 and 0.0 denote 100% and 0% of all sequences within cluster O and other KSB, respectively. *Unpublished data.

**Table 4 viruses-11-00545-t004:** Comparison of sequence identity with donors O and P within each cluster.

Gene	Cluster O (*n* = 9)	Cluster P (*n* = 13)	KSB (*n* = 19) [13]
*pol*	99.1 ± 0.5	99.2 ± 0.4^a^	97.1
*gag*	98.9 ± 0.2	98.9 ± 0.5	95.2
*vif*	98.9 ± 0.5	98.9 ± 0.7	96.2
*nef*	97.6 ± 1.6	97.3 ± 2.1^a, b^	95.6
*env*	96.3 ± 1.7	94.5 ± 3.2^b^	89.8

^a^*p* < 0.01 and ^b^
*p* < 0.05 in cluster P only. *p* < 0.05 between the *gag* and *nef* genes and *p* < 0.01 between the *gag* and *env* genes in both clusters. *p* < 0.05 between the *vif* and *env* genes in cluster O and between the *vif* and *nef* genes in both clusters. The sequence identity of clusters O and P in all genes was higher than that of published KSB (*n* = 19) by ≥2%.

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
