# Peer review of "Genetic Analysis of the Full-Length gag Gene from the Earliest Korean Subclade B of HIV-1: An Outbreak among Korean Hemophiliacs"

_viruses, 2019, doi:10.3390/v11060545_

Round 1
Reviewer 1 Report
I regret that the authors did not answered to all my (maybe too curious) comments. However, the main issues have been corrected.
I think that the manuscript has been reasonably improved and is suitable for publication.
Author Response
Thank you.
We significantly improved the manuscript by accepting academic editor’s excellent comments.
Reviewer 2 Report
The authors responded properly to the reviewers' comments and revised the manuscript adequately.
Author Response
Thank you very much.
This manuscript is a resubmission of an earlier submission. The following is a list of the peer review reports and author responses from that submission.
Round 1
Reviewer 1 Report
In this work the authors provide a genetic analysis of the gag gene from an already well characterized subtype B cluster from South Korea. The manuscript is clear, even if some aspects can be improved, and is technically sounding. The study does not provide a huge novelty regarding the cluster description but findings are illustrative of phylogenetic limits with long-time lagged sequences. These results are also briefly compared with those observed with the other HIV genes in the previous published studies conducted with this cluster. However, this part of the study may have been deeper analysed in my opinion. The authors stated that bootstrap values were not given as below 75%. In my opinion a bootstrap value at 70% is already strong, especially in this context and I would have been interested to have an idea of the branch support for clusters O and P most recent common ancestors. I also wonder if some other methods, such has SH-like test or ultrafast bootstrap implemented in IQtree would have performed better. More importantly, as the authors discussed about the various strength of the results obtained with several HIV genes, and stated that the pol gene appeared to be the best to use with this dataset, I wonder what would be the strength of a near full genome analysis. Does the authors obtained near full genome for most of the strains across these various studies?
Material and Methods. Line 57. The various populations used in the study should be more clearly defined, named and explained. It is sometimes confusing across the manuscript. All Korean patients infected with the Korean subtype B strains are called KSB-infected patients, the other Korean patients infected with a subtype B are mostly called subtype B infected patients throughout the manuscript. They should be called non-KSB-infected patients as subtype B can also referred to non-KSB populations and all other subtype B strains collected outside Korea. The description of the global Korean population is given in the Result section (line 105) but should appeared in the method section. The samples used should also be deeper explained here, it was not clear to me that all available samples were tested for each included patient.
Results. Line 119. The sentence should be written “In particular, THE two sequences from donor O strongly clustered…” and “In contrast THE two sequences from donor P clustered…”. Without “THE” it may suggest that if two sequences clustered, some others exist elsewhere in the tree.
Results. Table 1. The authors should replace “Background” by “Other KSB”, as background may refer to other KSB only or also including non-KSB sequences.
Results. Table 2. Subtype B should be replaced by non-KSB as subtype B can refer to non-KSB alone or also including the other subtype B strains included in some analysis of the study.
Results. Line 162. The authors provide in this paragraph interesting data about a duplication observed among some patients. Does this duplication also exist among the LANL database sequences? The authors also stated that the patient HjHi, presenting such duplications, was a long term non-progressor. They also stated that the clinical course of all other patients are depicted in the previous studies. A quick word stating the clinical status of the other patients should be added to help the reader without going through previous studies.
Results. Figure 2. The authors may add a shape to each point to distinguish those coming from the cluster O or P.
Conclusion. Line 247. The authors conclude that pol is the most appropriate when long-time lagged samples are included. However, this should be previously thoroughly discussed in the discussion and it was not clear to me which data allow this conclusion. I suppose that this is because of the higher number of signature pattern nucleotide identified here than in gag. But then, this may only be explained by the higher length of pol and not by evolution in pol. Moreover, the env gene seems also well appropriate to me according table 3. The authors also conclude that signature pattern analysis is more important than phylogenetic analysis. Again, I do not see the data supporting this assessment. If the branch support values are poor and if signature pattern analysis is highly valuable to confirm the links between sequences, I do not see where the phylogenetic tree presented in this study failed or provided wrong conclusions?
Reviewer 2 Report
In this report, Young-Keol Cho and co-workers studied the earliest full-length HIV-1 gag gene sequences derived from 110 Korean HIV-1-infected patients. The gag sequences were classified into 94 KSB and 16 subtype B sequences by a phylogenetic tree analysis. In addition, twelve signature pattern of amino acids were identified among KSB viruses and those were distinct from consensus subtype B samples. Moreover, there were 2 clusters of viruses found among 20 hemophiliacs samples, and 14 signature nucleotides were also found among 20 hemophiliacs and 90 local control samples. The earliest Korean full-length gag sequences were studied in this report, and it may provide valuable information in order to reveal the pre-epidemic situation of Korean HIV-1 at least in part. However, there are some questions remain to be addressed and modification is desirable in order to improve the manuscript. Specific comments are as follows.
1. It is known that subtype B viruses are prevalent world-wide, and are classified into several clades, according to a recent review article (Virology 495, 173-184, 2016). Beside 94 sequences belong to KSB, which subtype B clade the remaining 16 sequences belong to?
2. It is mentioned in the Introduction section that signature pattern analyses had done in previous reports; however, it is quite difficult to understand the analysis without reading the previous reports. I think it is necessary to explain the signature pattern analysis more in detail. In addition, it is also mentioned that among various HIV-1 genes, pol gene is the most suitable one for the signature pattern analysis; however, it had done on gag genes in this report. Please add more discussion to convince why the authors conduced the analysis on gag but not pol genes.
3. It is not clear what the values in Table 1 indicate. It is better explain more in detail. Is it considerable that particular nucleotide signature patterns were found 100% of clusters O and P, but were found quite rarely among background group? Did a value, 0.04 on the Table mean that 4 in 100 samples showed the pattern? If so, were the particular signature patterns found only among 9 cluster O and 13 cluster P samples? Was the nucleotide signature pattern found by comparing KSB and world-wide subtype B consensus sequences? Please explain more in detail. The comments are also applicable for Table 3 too.
4. There are 96 KSB gag sequences studied in this report; however, totally 41 (9+13+19) samples were shown in Table 4. I think the similarities of genes isolated in a particular year were studied in Table 4, but please explain more in detail.
5. Subtype B viruses are widely prevalent in other Asian countries such as Japan, the Philippines, Myanmar, Malaysia, China and Singapore. If the amino acids signature pattern among viruses prevalent in such countries will be studied in a future, is it possible to reveal possible transmission dynamics of HIV-1 infection among Asian countries at least in part ? In addition, has KSB viruses achieved their own evolution? What is the driving force? Is the amino acids signature pattern analysis useful to find the answers? Please add some discussion in order to show the importance of the analysis.